



# Technical note: Improved handling of potential evapotranspiration in hydrological studies with *PyEt*

Matevz Vremec[1], Raoul A. Collenteur[2], and Steffen Birk[1]

[1]Institute of Earth Sciences, NAWI Graz Geocenter, University of Graz, Graz 8010, Austria
[2]Eawag, Swiss Federal Institute of Aquatic Science and Technology, Department Water Resources and Drinking Water, Überlandstrasse 133, 8600 Dübendorf, Switzerland

**Correspondence:** Matevz Vremec (matevz.vremec@uni-graz.at)

**Abstract.**

Evapotranspiration (ET) is a crucial flux of the hydrological water balance, commonly estimated using (semi-)empirical formulas. The estimated flux may strongly depend on the used formula, adding uncertainty to the outcomes of hydrological models using ET. Climate change may cause additional uncertainty as each formula may respond differently to changes in

meteorological input data. To include the effects of model uncertainty and climate change, and facilitate the use of these formulas in a consistent, tested, and reproducible workflow, we present *PyEt*. *PyEt* is an open-source Python package for the estimation of daily potential evapotranspiration (PET) using available meteorological data. It allows the application of twenty different PET methods on both time series (Pandas) and gridded datasets (xarray). Most of the implemented methods are benchmarked against literature values and tested with continuous integration to ensure the correctness of the implementation.

This article provides an overview of *PyEt*'s capabilities, including the estimation of PET with twenty PET methods for station, and gridded data, a simple procedure for calibrating the empirical coefficients in the alternative PET methods, and estimation of PET under warming and elevated atmospheric $CO_2$ concentration. Further discussion on the advantages of using *PyEt* estimates as input for hydrological models, sensitivity/uncertainty analyses, and hind/forecasting studies, especially in data-scarce regions, is provided.

## 1 Introduction

Evaporation is the process where a substance (here, water) is converted from a liquid into a vapor phase. The evaporative flux is one of the major fluxes in the global hydrological cycle (Katul et al., 2012), and has a large impact on both human societies and ecosystems (Oki and Kanae, 2006; Fisher et al., 2011). A considerable part of this flux occurs through the root water uptake and transpiration of plants and thus is affected by plant physiological processes. In the remainder of this paper, the term

evapotranspiration is used to refer to the total evaporation flux from a land surface, consisting of transpiration (evaporation of water by vegetation), soil evaporation, and interception evaporation (Miralles et al., 2020). Evapotranspiration reduces the amount of water available to recharge and replenish groundwater resources, and determines how much water is needed for irrigation to ensure efficient and sufficient food production (e.g., Allen et al., 1998; Jensen and Allen, 2016). As such, accurate estimation of this flux is of paramount importance in hydrology, the geosciences and related fields.





Evapotranspiration (ET) can hardly be measured directly (Wang and Dickinson, 2012; Jensen and Allen, 2016), and is therefore commonly estimated using (semi-)empirical formulas from other, more easily obtained meteorological variables such as temperature, wind speed, and radiation. Over time, dozens of methods have been proposed and applied. Each of these methods generally results in slightly different estimates of evapotranspiration, depending on the methods and data used (Oudin et al., 2005; McMahon et al., 2013; Xu and Singh, 2000, 2001; Lemaitre-Basset et al., 2022). Most of these formulas estimate
either the reference crop evapotranspiration ($ET_0$), which is ET from a reference surface or crop that is not short of water (Allen et al., 1998), or the potential evapotranspiration (PET), which is the maximum rate of ET that would occur given a sufficient water supply (Xiang et al., 2020).

Potential evapotranspiration is determined by meteorological conditions, whereas water availability determines if actual evapotranspiration occurs at its potential rate (Jensen and Allen, 2016). Differences in the potential evapotranspiration may
cascade through a modeling chain and ultimately impact the results of a study. For example, Prudhomme and Williamson (2013); Lemaitre-Basset et al. (2022); Bormann (2010) showed that the method used affects the results from hydrological climate change impact studies. Similarly, the estimation of water demand for efficient crop and irrigation management depends on potential evapotranspiration, and may thus be impacted by the methods used (Kumar et al., 2012).

To account for the structural uncertainty of the different PET models, it has been recommended to use multiple methods
(Seiller and Anctil, 2016; Beven and Freer, 2001; Velázquez et al., 2013). Such an approach can help improve the understanding of the effect of model uncertainty on PET estimates in, for example, historical climate studies (Zhou et al., 2020; Dakhlaoui et al., 2020; Yang et al., 2019) and climate change impact studies (Bormann, 2010; Seiller and Anctil, 2016; Gharbia et al., 2018; Shi et al., 2020). It may also be necessary to account for environmental variables that change over time and impact the evapotranspiration, such as vegetation changes and increases in atmospheric $CO_2$ concentrations (Fatichi et al., 2016;
Ainsworth and Rogers, 2007; Vremec et al., 2022). To efficiently account for the structural uncertainty, the software used to compute PET should ideally have multiple methods available, and be flexible enough to deal with such changing environment variables.

In practice it is a common approach to use high-level open-source programming languages (e.g., R, Python, Julia) and scripts for the estimation of PET. Various open-source libraries exists for common tasks such as PET estimation in different languages.
To *R* Programmers, the package 'Evapotranspiration' (Guo et al., 2016) provides a community library with many different PET methods. In the Python community, development is more diffuse and several packages exist that implement one or a few PET methods (e.g., Richards, 2019; Kittridge, 2019; Morton, 2020; Christofides, 2020). Even combined, however, these packages do not nearly provide as many PET methods as the R package. Given the widespread use of the Python programming language and the common reliance on PET estimation methods in the geosciences (and beyond), we argue that the scientific community
would benefit from a single Python package for the estimation of PET, that implements many PET estimation methods, and importantly, is well-documented and tested.

In this paper we introduce *PyEt*, an open-source Python package for the estimation of potential evapotranspiration. The aim of *PyEt* is to provide researchers and practitioners with a wide variety of tested, documented, and flexible Python functions for the estimation of potential evapotranspiration. All methods have a common application programming interface, allowing





users to easily test different PET models for their application and, if desired, address structural uncertainty and changing
conditions. The majority of the implemented methods are benchmarked against literature values and tested with continuous
integration to ensure the correctness of the implementation. Allowing different types of input data, *PyEt* is also applicable in
regions with sparsely distributed measurement stations, where standard meteorological data (e.g., wind, relative humidity) are
often unavailable. The software is available under MIT-licence from the Python Package Index (PyPI) (Vremec and Collenteur,
2022), and developed as a community project on Github (www.github.com/pyet-org/PyEt).

The remainder of this paper is structured as follows. In the next section the software design, capabilities, and benchmarking
tests are described. The third section introduces the software through four examples showing potential future users how to
apply *PyEt* in real-world applications. In these examples, we focus on practical problems encountered in the everyday life
of hydroligsts. The fourth section discusses future potential applications of *PyEt*, and how we think it can help the scientific
community improve the estimation of potential evapotranspiration. In the fifth and final section, conclusions and future plans
are outlined.

## 2   PyEt Python Package

### 2.1   Software design

The basic design principle for *PyEt* was to built a software that is intuitive and easy-to-use by novice users with little pro-
gramming experience, yet flexible enough to allow advanced users to perform more complex analyses. The software uses a
modular design, with formulas used by different PET methods implemented as a single function. This reduces the amount of
code and makes it easier to maintain the software and implement new methods. All the PET methods are intended to work
with the minimum input data required by the PET models (e.g., radiation, temperature), but also allow more user input if such
data is available and allowed by the PET method (e.g., humidity, surface resistance in the Penman-Monteith model). If data is
unavailable, utility functions are available to the user or are called internally to compute the unavailable variables (e.g., solar
radiation from latitude value). Moreover, the constants in the empirical PET formulas are function arguments with default
values from the literature, which may also be changed by the user to adapt the empirical relationship to another region. Finally,
the available methods should work for both station and gridded data.

*PyEt* is part of the wider Python ecosystem, and depends on three widely used and well-developed Python packages from the
Scientific Python stack: Numpy (Harris et al., 2020), xarray (Hoyer and Hamman, 2017), and Pandas (McKinney, 201). The
input and output of *PyEt* are formatted as time series data in Pandas.Series or xarray.DataArrays, which allows to use all of the
Pandas/xarray functions on the data (Harris et al., 2020; McKinney, 201; Hoyer and Hamman, 2017). These functions include
gap-filling and selection functions for interpolation, resampling, clustering, and many more. Being part of a wider ecosystem,
users can leverage other Python packages for visualisation (e.g., Matplotlib (Hunter, 2007), MetPy (May et al., 2022)) and
optimization and uncertainty analyses (Scipy (Virtanen et al., 2020), SpotPy (Houska et al., 2015)).

The software is hosted and developed on the GitHub platform, and distributed under MIT-license through the Python
Packaging Index (PyPI). Documentation and example applications are available on a dedicated ReadTheDocs website (http:





//PyEt.readthedocs.io). The documentation for individual methods is also directly available in Python from the documentation strings. Each release of *PyEt* is automatically stored in the Zenodo repository and assigned a Digital Object Identifier (DOI).

As such, *PyEt* complies with many of the recommendations for good research software development as given in, for example, Hutton et al. (2016) and the FAIR4RS principles (Barker et al., 2022). The scripts or the Jupyter notebooks used to apply *PyEt* provide full reproducibility and a transparent report of the entire calculation process (Kluyver et al., 2016).

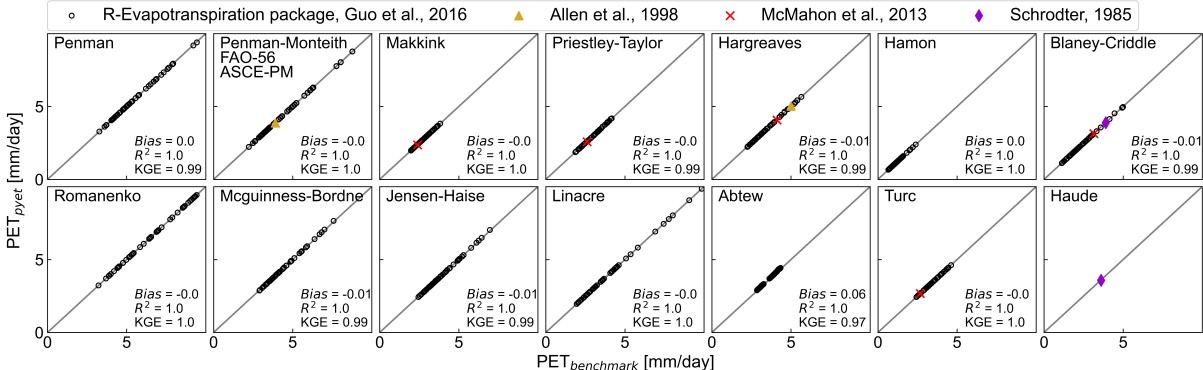

**Figure 1.** Scatter plots showing estimated PET with *PyEt* against PET values estimated with the R package $Evapotranspiration$ from Guo et al. (2016), and literature values from Allen et al. (1998), McMahon et al. (2013), and Schrödter (1985).

## 2.2 Implemented methods and benchmarking

Twenty methods are currently implemented in *PyEt* for the estimation of daily potential evapotranspiration. This includes the
most popular methods such as Penman-Monteith, Hamon, and Hargreaves. An overview of these methods and the required input data is provided in Table 1. Depending on the method, different (amounts of) input data are required to compute the potential evapotranspiration. It is often also possible to provide different input data to the same method (e.g., the average or the minimum and maximum daily temperatures), or even that some of the input data is optional (as described in the footnotes of Table 1). In the case of optional input data, utility functions are used internally to estimate that data. In the example of the
Penman-Monteith, solar radiation does not necessarily need to be provided by the user and can be estimated from the latitude and actual duration of sunshine hours instead.

The *PyEt* project is intended to be used by a wide community and any errors in the code may have consequences for other studies applying *PyEt* to obtain PET estimates. Special attention was therefore paid to benchmark the available methods to published literature values and data from well-known research and meteorological institutes (Allen et al., 1998; McMahon
et al., 2013; Schrödter, 1985; Walter et al., 2000). These benchmarks are also implemented in the continuous integration and tested using the *unittest* testing framework (unittest, 2022). This ensures that the benchmarks are satisfied each time the software is updated in the future. New methods added to *PyEt* require the provision of benchmark data en tests. Figure 1 shows the results for each benchmarked method, indicating that the PET estimates from all these methods are equal to the benchmark



**Table 1.** Data requirements for different PET or $ET_0$ models, the corresponding *PyEt* function, and if benchmarking of the method was performed. Adapted after Guo et al. (2016); Oudin et al. (2005); McMahon et al. (2013).

| Method name[0] | *PyEt* function | Climate data | | | | Location | | Benchmark |
|---|---|---|---|---|---|---|---|---|
| | | $T$ | $RH$ | $R$ | $u_2$ | Lat. | El. | |
| PET: Penman | *penman* | ✓[a] | ✓[b,c] | ✓[d] | ✓ | ✓[d] | ✓[e] | ✓ |
| PET: Penman-Monteith | *pm* | ✓[a] | ✓[b,c] | ✓[d] | ✓ | ✓[d] | ✓[e] | ✓ |
| $ET_0$: ASCE-PM | *pm_asce* | ✓[a] | ✓[b,c] | ✓[d] | ✓ | ✓[d] | ✓[e] | ✓ |
| $ET_0$: FAO-56 | *pm_fao56* | ✓[a] | ✓[b,c] | ✓[d] | ✓ | ✓[d] | ✓[e] | ✓ |
| PET: Priestley-Taylor | *priestley_taylor* | ✓ | ✓[h] | ✓[h] | - | ✓[h] | ✓[e] | ✓ |
| PET: Kimberly-Penman | *kimberly_penman* | ✓[a] | ✓[b,c] | ✓[d] | ✓ | ✓[d] | ✓[e] | - |
| PET: Thom-Oliver | *thom_oliver* | ✓[a] | ✓[b,c] | ✓[d] | ✓ | ✓[d] | ✓[e] | - |
| PET: Blaney–Criddle | *blaney_criddle* | ✓ | -[i] | -[i] | -[i] | ✓ | - | ✓ |
| PET: Hamon | *hamon* | ✓ | - | - | - | ✓ | - | ✓ |
| PET: Romanenko | *romanenko* | ✓ | ✓ | - | - | - | - | ✓ |
| PET: Linacre | *linacre* | ✓[j] | - | - | - | - | ✓ | ✓ |
| PET: Haude | *haude* | ✓ | ✓[k] | - | - | - | - | ✓ |
| PET: Turc | *turc* | ✓ | ✓ | ✓ | - | - | - | ✓ |
| PET: Jensen–Haise | *jensen_haise* | ✓ | - | ✓[l] | - | ✓[l] | - | ✓ |
| PET: McGuinness–Bordne | *mcguinness_bordne* | ✓ | - | - | - | ✓ | - | ✓ |
| PET: Hargreaves | *hargreaves* | ✓[m] | - | - | - | ✓ | - | ✓ |
| $ET_0$: FAO-24 | *fao_24* | ✓ | ✓ | ✓ | ✓ | - | ✓[e] | - |
| $ET_0$: Abtew | *abtew* | ✓ | - | ✓ | - | - | - | ✓ |
| PET: Makkink | *makkink* | ✓ | - | ✓ | - | - | ✓[e] | ✓ |
| PET: Oudin | *oudin* | ✓ | - | - | - | ✓ | - | - |

[0] The corresponding literature to each method is provided in Table A1, in Appendix. [a] $T_{max}$ and $T_{min}$ can also be provided. [b] $RH_{max}$ and $RH_{min}$ can also be provided. [c] If actual vapor pressure is provided, RH is not needed. [d] Input for radiation can be (1) Net radiation, (2) solar radiation or (3) sunshine hours. If (1), then latitude is not needed. If (1, 3) latitude and elevation is needed. [e] One must provide either the atmospheric pressure or elevation. [f] The PM method can be used to estimate potential crop evapotranspiration, if leaf area index or crop height data is available. [g] The effect of $CO_2$ on stomatal resistance can be included using the formulation of Yang et al. 2019. (Yang et al., 2019). [h] If net radiation is provided, RH and Lat are not needed. [i] If method==2, $u_2$, $RH_{min}$ and sunshine hours are required. [j] Additional input of $T_{max}$ and $T_{min}$, or $T_{dew}$. [k] Input can be RH or actual vapor pressure. [l] If method==1, latitude is needed instead of $R_s$. [m] $T_{max}$ and $T_{min}$ also needed.

values. Despite our best efforts, we acknowledge here that a few methods have not yet been benchmarked due to a lack of

appropriate data.


## 2.3 The Penman-Monteith equation

For illustrative purposes for the remainder of this paper we discuss one of the most used and versatile PET methods in more detail here: the Penman-Monteith method (Monteith, 1965). In its different forms, the Penmam-Monteith equation can be used to estimate reference crop evapotranspiration (Allen et al., 1998; Walter et al., 2000), potential evapotranspiration (Monteith, 1965), and potential crop evapotranspiration ("the maximum value of ET from a specific crop type having specific properties under conditions of full soil water supply, but not necessarily having a saturated surface"; Jensen and Allen (2016)). Monteith (1965) enabled the application of the Penman-Monteith equation to a wide range of surfaces and vegetation types (Jensen and Allen, 2016), by implementing the plant aerodynamic resistance ($r_a$) and the surface resistance ($r_s$) in the formula.

Users of *PyEt* can include leaf/canopy cover measurements (Leaf Area Index - LAI) to calculate surface resistance ($r_s$), thereby accounting for the effects of crop management and phenology on ET. A modified stomatal resistance model also allows for the inclusion of the sensitivity of the stomatal resistance ($r_l$) to the atmospheric $CO_2$ concentration (e.g., Yang et al., 2019; Vremec et al., 2022):

$$r_s = \frac{r_l(CO_2)}{0.5LAI} = \frac{r_{r_l-300} \times \left\{ 1 + S_{r_l-CO_2} \times (CO_2 - 300) \right\}}{0.5LAI} \tag{1}$$

where $S_{r_l-[CO_2]}$ [ppm$^{-1}$] is the relative sensitivity of $r_l$ to $\Delta$ [$CO_2$] and $r_{r_l-300}$ [s m$^{-1}$] is the reference stomatal resistance when atmospheric $CO_2$ concentration is 300 ppm. The relative sensitivity of $r_l$ to $\Delta$ [$CO_2$] represents the change in $r_l$ per ppm increase in $CO_2$ concentration.

If measurements of the crop height exist, these data can be used in *PyEt* to calculate the aerodynamic resistance to vapor and heat transfer ($r_a$) to better represent the effects of crop phenology on PET:

$$r_a = \frac{ln\left[\frac{z_m-d}{z_{om}}\right] ln\left[\frac{z_h-d}{z_{oh}}\right]}{k^2 u_z} \tag{2}$$

where $z_m$ is the reference level at which the wind speed is measured; $z_h$ is the height of the temperature and humidity measurements; $k$ is the von Karman constant ($= 0.41$), $u_z$ is the measured wind speed (Allen et al., 1998) and $d$ is the zero plane displacement height, taken as $0.67h_c$; $z_{om}$ is the roughness parameter for momentum ($= 0.123h_c$) and $z_{oh}$ is the roughness parameter for heat and water vapor ($= 0.1z_{om}$) (Jensen and Allen, 2016).

## 3 Example use cases

Below we present four example use cases of *PyEt* to illustrate how the software can be used and for what types of analyses it can be applied. The first example shows how to efficiently compute different potential evapotranspiration estimates using 20 various methods for station data. This example also illustrates how to use *PyEt* in general. The second example illustrates how to provide 3D estimates of PET using 3 different methods and gridded xarray data. The third example shows how to calibrate





different PET methods to local conditions and use the calibrated formula for hindcasting. The fourth example illustrates a
145  workflow to account for the effects of warming and elevated $CO_2$ in climate change impact studies. The source code for these
and other examples can be found in the 'examples' folder of the GitHub website of the project.

### 3.1 Example 1: Estimation of PET from station data

In this example potential evapotranspiration is estimated for the town of De Bilt in The Netherlands using data provided
by the Royal Netherlands Meteorological Institute (KNMI). The reference method used by the KNMI for the estimation of
150  potential evapotranspiration is the Makkink method, also implemented in *PyEt*. The PET computed with the Makkink method
is compared to the PET values from all other methods in *PyEt*. A number of steps are taken in a Python script to estimate PET.
The code implementing these steps is shown in the code example bellow. *PyEt* provides a convenience method to compute the
PET with all available methods, *pyet.calculate_all()*:

1. Import the necessary Python packages.

```
import pandas as pd
import pyet
```

2. Load the meteorological data.

```
meteo = pd.read_csv("meteo.csv",
   index_col=0, parse_dates=True)
```

3. Determine the necessary input data for the PET model.

```
tmean, tmax, tmin, rh, rs, wind, \
pet_knmi = (meteo[col] for col in
           meteo.columns)
lat = 0.91  # define latitude
elev = 4  # define elevation
```

4. Estimate the potential evapotranspiration with all methods or the method of choice.

```
pet_df = pyet.calculate_all(tmean,
   wind, rs, elev, lat, tmax, tmin, rh)
pet_mak = pyet.makkink(tmean, rs,
   elevation=elev)
```

5. Visualize and analyze the results.





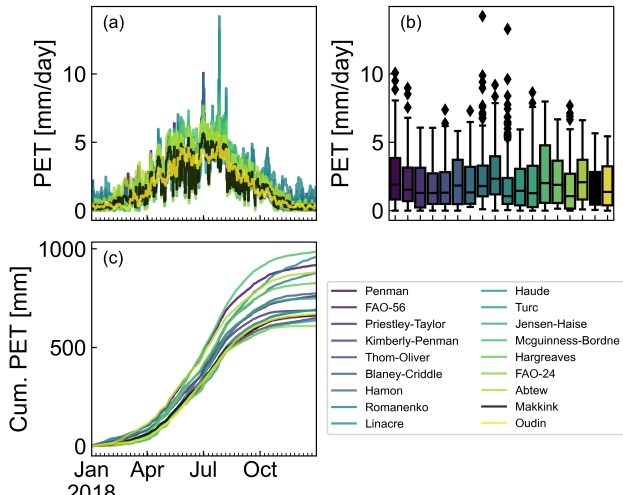

**Figure 2.** Computed potential evapotranspiration estimates plotted as time series (a), box plots (b), cumulative PET (c).

```
pet_df.plot()
pet_df.boxplot()
pet_df.cumsum().plot()
```

The results from the above analysis are shown in Figure 2. From these visualizations, it is clear that the potential evapo-transpiration depends on the chosen method, which may accumulate up to a 35% deviation of the estimated annual flux from the mean in this example. Such substantial differences between the estimated fluxes motivate the use of multiple methods (ensemble modelling) (Beven and Freer, 2001; Krueger et al., 2010; Shi et al., 2020; Oudin et al., 2005). This example shows how *PyEt* can be used efficiently for this task, without much additional effort or many lines of code.

**3.2    Example 2: Estimate PET for gridded data**

Although time series data is probably still the most commonly available data format, gridded 3-dimensional data (x,y,t) obtained from satellites, radar imagery, or post-processed products is rapidly becoming widely available. More and more public data sets exist with global PET estimates at 0.1 degree resolution (e.g., Martens et al., 2017; Xie et al., 2022), providing valuable input data for many studies. *PyEt* also supports such gridded data, as illustrated here for the E-OBS gridded data (Cornes

et al., 2018) for Europe. The application of *PyEt* on gridded datasets is displayed using the FAO-56, Makkink, and Hargreaves methods. Instead of Pandas Series as the input data type for the PET method, now xarray.DataArrays are used as the input data type. *PyEt* methods will return the same data type, again a xarray.DataArray. The workflow is the same as in the first example, except that we will now evaluate the PET for each method separately.

The results for the three methods and three time steps are shown in Figure 3. These again show that, depending on the PET

method, results may differ, also spatially. Looking more closely at Figure 3, we can observe that the FAO-56 and Makkink

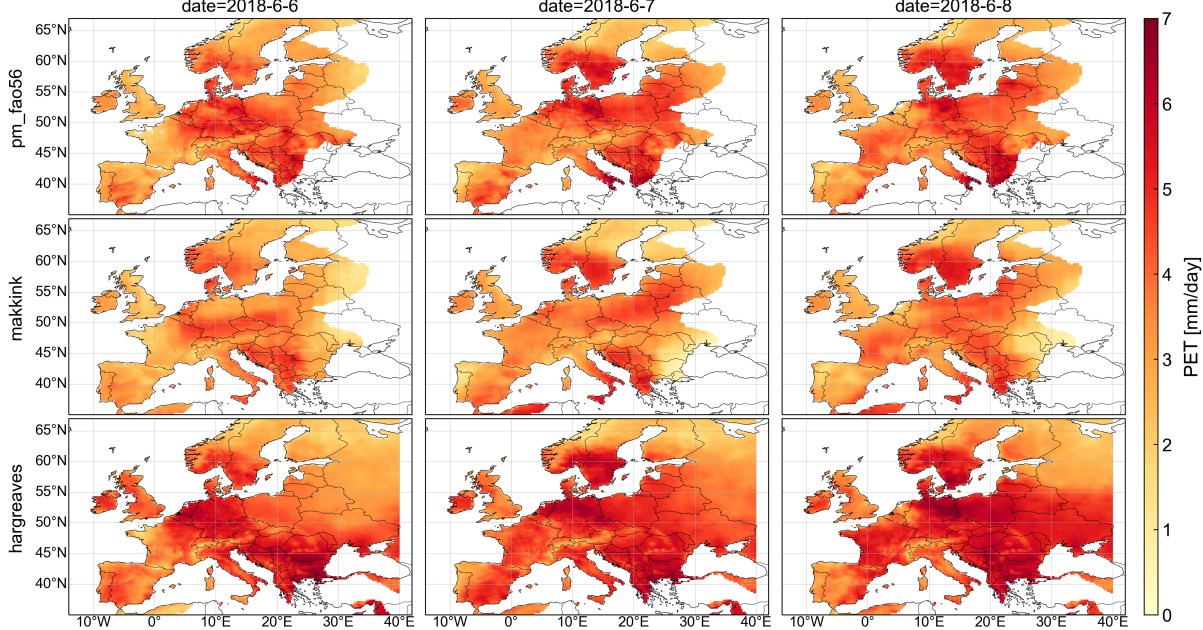

**Figure 3.** Daily PET estimates for Europe from 2018-6-6 to 2018-6-8 using meteorological data obtained from the E-OBS dataset (Cornes et al., 2018).

method do not compute PET in eastern parts of Europe. These areas do not include relative humidity and solar radiation data, thus PET cannot be computed using the FAO-56 or Makkink method. If NaN (not-a-number) values are present in the required input data for a *PyEt* method, the method also returns a NaN value. On the other hand, the Hargreaves method does not require solar radiation or relative humidity data, so it can compute PET in the eastern parts of Europe. This example demonstrates how

*PyEt* can be applied to estimate PET using gridded data and demonstrates the benefits of using alternative PET methods when radiation, wind or relative humidity data are missing.

### 3.3   Example 3: Calibration of PET models

The available input data often does not suffice to compute potential evapotranspiration with the Penman-Monteith equation. This can be the case in data-scarce regions or time periods, or when using historical data or data from climate models. In such

cases, one can calibrate alternative PET methods to the estimates from the Penman-Monteith equation for the period when enough data is available. The calibrated method can than be used to estimate PET in the period of data scarcity. As concluded by several authors (Jensen and Allen, 2016; Valipour, 2015; Yang et al., 2021; Dlouhá et al., 2021), calibration of alternative models is often crucial to ensure that the model fits the regional climate. In this example, we show how the calibration of temperature-based PET models affects the model uncertainty for studies focusing on current and past climates.





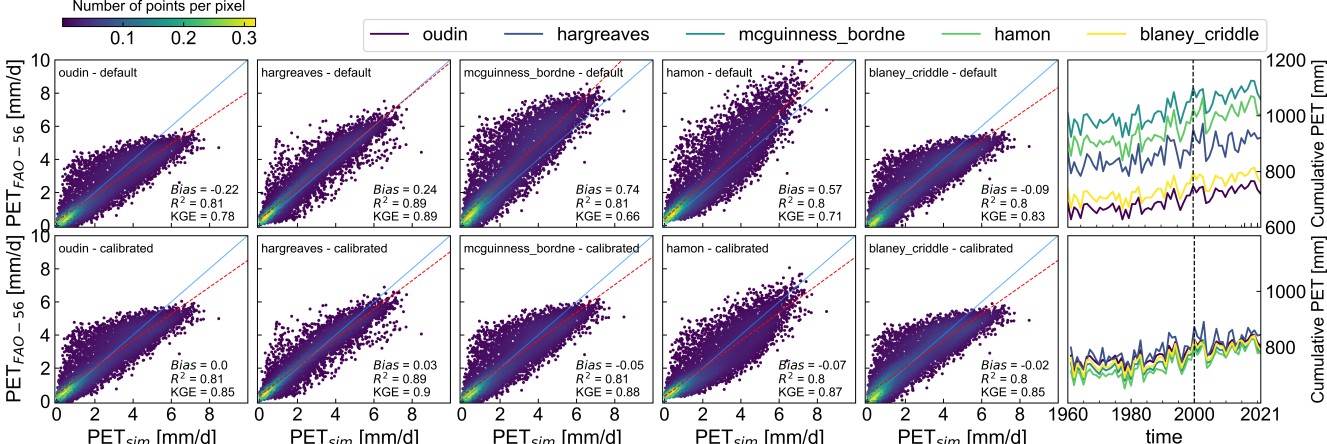

**Figure 4.** Density scatter plots comparing simulated and observed (FAO-56) PET for uncalibrated (row 1) and calibrated models. Last column shows the aggregated annual PET for the period 1961-2021 using uncalibrated (row 1) and calibrated models (row 2).

The approach is illustrated for the regions of Graz, Austria, where the input data required for Penman-Monteith are only available from 2000 to 2021. Imagine, however, that for our study we also need potential evapotranspiration data for the period 1961 to 2021, but only temperature data is available (e.g., from the Spartacus temperature dataset (Hiebl and Frei, 2016)). A number of steps are taken to calibrate the following five temperature-based methods: Oudin, Hargreaves, McGuiness-Bordne, Hamon, and Blaney-Criddle. First, the PET for the period 2000-2021 is computed using the Penman-Monteith equation. In

the second step, the coefficients of the temperature-based PET equations are estimated by calibrating the estimated PET from temperature-based methods to the Penman-Monteith PET. Calibration is done by minimizing the sum of the squared residuals between these two PET estimates, using SciPy's (Virtanen et al., 2020) *least_squares* method. In the third and final step, these calibrated coefficients are used to estimate the PET for the period 1961-2021.

     Figure 4 shows the computed PET with default (row 1) and the calibrated coefficients (row 2). The model bias (mm/day)

and the coefficient of determination ($R^2$) between simulated and observed (Penman-Monteith) PET show an improved model fit for all methods after calibration. The use of calibrated methods reduces the model bias, which is visually illustrated by the annual PET flux (composed of daily values) in the last column of Figure 4. Using the Spartacus temperature dataset (Hiebl and Frei, 2016), we can now estimate PET up to 1961 using the calibrated alternative PET methods.

## 3.4    Example 4: The effect of CO$_2$ on future PET estimates

In this example, it is shown how to account for changing environmental conditions affecting the PET flux when modelling the effects of climate change. Under a warmer and CO$_2$ richer future (Caretta et al., 2022), potential evapotranspiration tends to increase with increasing temperature (and vapor pressure deficit), while a reduction in PET is expected under elevated CO$_2$ due to an increased stomatal resistance (Field et al., 1995; Ainsworth and Rogers, 2007). The increase in CO$_2$ is still





commonly ignored in PET models employed for climate change studies, although excluding its stomatal effect may lead to an
overestimation of PET (Kingston et al., 2009; Milly and Dunne, 2016; Vremec et al., 2022). The increase in temperature can be
easily modelled with all available PET methods, as temperature is an input for all methods, while the $CO_2$ stomatal effect can
only be directly accounted for with the Penman-Monteith method (Liu et al., 2022). Using a $CO_2$-dependent stomatal resistance
model implemented in *PyEt* (Yang et al., 2019), the effect of elevated $CO_2$ on stomatal resistance can be considered (see eq.
1). When calculating PET with alternative methods, Kruijt et al. (2008) and Trnka et al. (2014) argues that an adjustment
factor for the atmospheric $CO_2$ concentration ($f_{CO_2}$) can be used to account for the effect of elevated $CO_2$ concentrations on
PET. The scaling factor can be obtained from literature values (Kruijt et al., 2008; Trnka et al., 2014), or calibrated using the
Penman-Monteith equation together with the $CO_2$-dependent stomatal resistance model (eq. 1) to match the regional climate
and vegetation:

$$\mathrm{PET_{CO_2}} = f_{\mathrm{CO_2}}\mathrm{PET_{300}}$$
$$= (1 + S_{\mathrm{PET_{CO_2}}}(\mathrm{CO_2} - 300))\mathrm{PET_{300}} \qquad (3)$$

where $S_{PET_{CO_2}}$ is the relative sensitivity of PET to $CO_2$, $\mathrm{PET_{300}}$ is the computed Penman-Monteith estimate at 300ppm [$CO_2$]
(preindustrial concentration), while $\mathrm{PET_{CO_2}}$ is the computed Penman-Monteith estimate under elevated $CO_2$ concentration
(Yang et al., 2019). Such relationship can be easily implemented in *PyEt*, and $f_{CO_2}$ can be obtained by calculating $\mathrm{PET_{300}}$ and
$\mathrm{PET_{CO_2}}$ with the Penman-Monteith equation (eq. 1) at ambient and elevated $CO_2$ concentration, respectively.

For our study region, we used the calibrated models from the previous example to analyse the effects of warming and elevated
$CO_2$ concentration on PET based on the projected increase in temperature and $CO_2$ concentration from the representative
concentration pathways (RCPs) (Van Vuuren et al., 2011). We calculated daily PET for each RCP scenario (2.6, 4.5, 6.0
and 8.5) by adding the projected increase in temperature and $CO_2$ concentration to the existing data for 2020-2021. Figure 5
shows the increase in the average annual PET (aggregated from daily values) under warming and elevated $CO_2$ concentrations
according to the RCP scenarios. In figure 5-c, the effects of elevated $CO_2$ concentration on PET were neglected, and only
increase in temperature was considered. Similar to Milly and Dunne (2016) , Yang et al. (2019) and Vremec et al. (2022), this
example shows that neglecting the effect of elevated $CO_2$ on PET (Fig, 5-c) can lead to overestimation of PET under future
conditions.



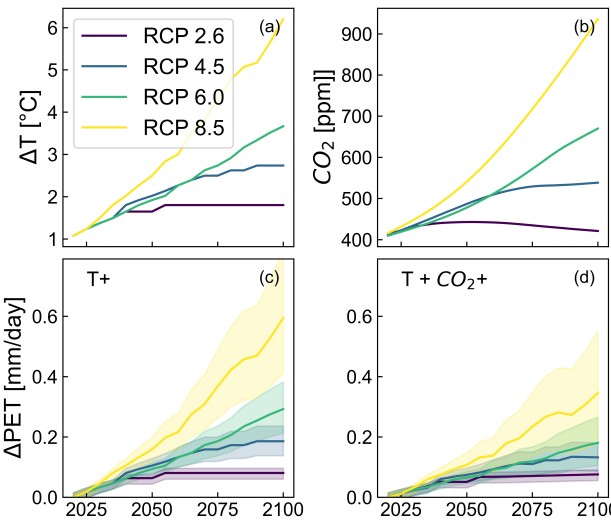

**Figure 5.** Projected increase in temperature (5-a) and atmospheric $CO_2$ concentration (5-b) under the RCP scenarios, and calculated increase in the average annual PET with warming (5-c), and PET with warming and elevated $CO_2$ concentration (5-d). The uncertainty bounds represent the 5th-95th percentile of the PET model ensemble.

## 4 Discussion

### 4.1 Improved handling of PET in scientific studies

Evapotranspiration data from lysimeter or eddy correlation measurements (Pastorello et al., 2020) are rare and, if available at all, only locally available for relatively short time periods. Thus, there is a need to estimate evapotranspiration from more readily available meteorological data using (semi-)empirical approaches. In general, these approaches comprise three steps, as outlined for example by Allen et al. (1998). Firstly, the potential evapotranspiration of a reference surface (hence reference evapotranspiration) is estimated using meteorological data. Secondly, a crop coefficient may be applied to transform the refer-

ence evapotranspiration into the potential crop evapotranspiration. Thirdly, a soil-water balance approach is used to account for reduced actual evapotranspiration if the soil-water storage is depleted. *PyEt* is designed to perform the first two steps. It can be easily complemented by soil-water balance approaches to calculate actual evapotranspiration. Hydrological models, however, often use PET directly as input.

Rainfall-runoff models represent one type of hydrological model where PET is commonly used as an input, either as a

gridded data in distributive models or as a spatially aggregated values in lumped-parameter models. Some studies (e.g., Andréassian et al., 2004; Oudin et al., 2005; Sperna Weiland et al., 2012) found that PET had little impact on the performance of such models and thus advocated the use of simplistic PET models. However, Jayathilake and Smith (2021) found that model performance was clearly sensitive to PET at sites where evapotranspiration was water limited. More importantly, the choice of the PET model has been shown to affect the results of hydrological projections in climate change impact assessments (Kay and





Davies, 2008; Seiller and Anctil, 2016; Dallaire et al., 2021; Lemaitre-Basset et al., 2022)). PET is expected to be even more influential in the assessment of groundwater recharge (e.g., Bakundukize et al., 2011) and crop water demands (e.g., Webber et al., 2016)), which – compared to runoff – are more directly linked to evapotranspiration. Thus, the selection of appropriate PET models needs to account for the research context and variable of interest. In addition, Bormann (2010) found that PET models that are based on the same or similar climate variables exhibit different sensitivity to observed climate change. This

finding suggests that appropriate PET models need to be specifically selected for the given region of interest. Guo et al. (2017) provides pointers on examining which variables are likely to be the most important for a particular location. The comparison of PET estimates for Europe shown in Figure 3 illustrates the differences in PET for different regions; as can be seen, the magnitude and pattern of PET estimates are similar in some regions (e.g., Scandinavia) but differ more strongly in others (e.g., Southeast Europe).

As indicated above, the performance of PET models may vary depending on the region considered. Thus, some approaches that were found to be applicable in other regions may perform less well in the region of interest. In this case, PET models can be calibrated to a reference data set by adjustment of the coefficients in the model equation as shown in the third example. The reference data set can either be observed evapotranspiration (e.g., from lysimeters) or PET obtained from a model considered to be reliable. This has been illustrated by Example 3, where the coefficients of temperature-based models were adjusted to

achieve the best fit to the Penman-Monteith model. This approach can also be used to obtain consistent spatial distributions of PET. As shown in Example 2 (Figure 3), the limited data availability for Eastern Europe did not allow the application of the FAO-56 or Makkink method, while sufficient data was available for the Hargreaves method. Thus, one may consider to calibrate the latter to one of the former methods where these are applicable and only then apply it to obtain estimates for the entire region. For a more advanced calibration procedure, see for example Haslinger and Bartsch (2016).

In many cases the range of PET models that can potentially be employed is pre-determined by data availability. This may be the case if historical records of climate data are to be used for the PET estimation, for example, as many weather stations do not measure all climate variables included in the Penman-Monteith equation. Yet, this is also often the case in assessments of hydrological impacts of climate change, if projected climate variables have high uncertainty. Lai et al. (2022), for example, concluded that the high uncertainty of wind speed projected in complex terrain may increase the uncertainty in PET, whereas

air temperature and solar radiation have low uncertainty and thus should be the parameters preferred in the PET model. Given the climate variables for which data is available, Table 1 can be used to identify the PET models that come into consideration. We generally recommend to apply all of the models for which data is available (PET model ensemble), but the purpose and specific implementation of such a multi-model approach will depend on the research context. Example 4 (section 3.4) further illustrated how PET model ensembles can be used to include model uncertainties in PET projections under warming

and elevated atmospheric $CO_2$ concentration. Since the latter effect is frequently excluded in hydrological projections, Milly and Dunne (2016) and Yang et al. (2019) advocate the inclusion of the effect of elevated $CO_2$ on stomatal resistance when estimating PET under warming and elevated atmospheric $CO_2$ concentrations.

To improve reliability and efficiency in estimating PET, it is crucial to allow a reproducible workflow. Scripts provide an efficient way to report on the modelling process and allow full reproducibility. As shown in the examples, Jupyter Notebooks





(Kluyver et al., 2016) provide a solution for publishing code, results, and explanations in a single document. As such, the presented package and its application in this paper are in line with the steps suggested by Hutton et al. (2016) to improve reproducibility in hydrological studies. To speed up adaptation of the methods and allow a faster transfer between research teams, formal procedures such as benchmarking (e.g., Maxwell et al., 2014) can help to ensure confidence in key complex codes.

## 4.2 Building the PyEt community and outlook

As a community project, the success of *PyEt* depends on the uptake from and interaction with the community. This, in turn, depends on the ease of use and the trust in the project. We therefore put a strong emphasis on designing a user-friendly programming interface with full documentation including various user examples, and extensive benchmark testing in the continuous integration. Since *PyEt* is available as a Python package, we have already seen a good community uptake and use of the pack-

age. Apart from applications of the software in projects related to the Authors (e.g., Vremec et al., 2022; Forstner et al., 2022; Collenteur et al., 2023), other independent researchers have successfully used *PyEt* in peer-reviewed studies (e.g., Vaz et al., 2022) and other unpublished works.

The primary channel for communication with the *PyEt* community is GitHub, which provides several options for discussions, tracking code issues, and code development. Users are encouraged to ask questions in GitHub discussions and to report

potential issues, suggest improvements, and feature requests via the GitHub issue tracker. As a community project, we hope to continue to improve the existing code and develop new capabilities based on feedback and with help from the community. An example of developments that are currently underway is the adaptation of the current methods to also work for hourly data, allowing the estimation of hourly PET. Other future work will focus on improvements in usability and the inclusion of other alternative methods.

## 5 Conclusions

In this paper we introduced *PyEt*, a Python package for the estimation of daily potential evapotranspiration (PET). The package enables the inclusion of model uncertainty and climate change into the estimation of PET in a consistent, tested, and reproducible environment. With *PyEt*, users can estimate PET using 20 different methods with only a few lines of Python code for both 1D (e.g. time series data) and 3D data (xarray). The examples described in this paper illustrate how *PyEt* can be employed

in hydrological studies to: (1) facilitate the characterisation of model uncertainty using a multi-model approach (model ensembles); (2) calibrate PET models and apply them in data-scarce regions and time periods; (3) include the effects of warming and elevated atmospheric $CO_2$ concentrations. The use of Python scripts and Jupyter Notebooks ensure reproducibility and provides a transparent report of the PET computation process. We believe that *PyEt* will improve the handling of PET and allow a more sophisticated and comprehensive consideration of PET in hydrological studies, particularly those related to climate

change.



**Table A1.** Corresponding literature for each method.

| Method name | Corresponding literature |
| --- | --- |
| PET: Penman | Penman (1948) |
| PET: Penman-Monteith | Monteith (1965) |
| $ET_0$: ASCE-PM | Walter et al. (2000) |
| $ET_0$: FAO-56 | Allen et al. (1998) |
| PET: Priestley-Taylor | Priestley and Taylor (1972); McMahon et al. (2013) |
| PET: Kimberly-Penman | Wright (1982) |
| PET: Thom-Oliver | Thom and Oliver (1977) |
| PET: Blaney–Criddle | Blaney and others (1952); Xu and Singh (2001); McMahon et al. (2013); Schrödter (1985) |
| PET: Hamon | Hamon (1963); Ansorge and Beran (2019); Oudin et al. (2005) |
| PET: Romanenko | Romanenko (1961); Xu and Singh (2001) |
| PET: Linacre | Linacre (1977) |
| PET: Haude | Haude (1955); Schiff (1975) |
| PET: Turc | Turc (1961); Xu and Singh (2001) |
| PET: Jensen–Haise | Jensen and Haise (1963); Jensen and Allen (2016); Oudin et al. (2005) |
| PET: McGuinness–Bordne | McGuinness and Bordne (1972) |
| PET: Hargreaves | Hargreaves and Samani (1982) |
| $ET_0$: FAO-24 | Jensen et al. (1990) |
| $ET_0$: Abtew | Abtew (1996) |
| PET: Makkink | Makkink (1957); McMahon et al. (2013) |
| PET: Oudin | Oudin et al. (2005) |

*Code and data availability.* The Jupyter Notebook and data used in this study are available in the "examples" folder of the GitHub repository and also available on Zenodo (version v.1.2.2, DOI: 10.5281/zenodo.5896799). The authors welcome code contributions, bug reports, and feedback from the community to further improve the software. *PyEt* is free and open-source software available under the MIT license. Source code is available at the project's home page on GitHub. Full documentation is available on ReadTheDocs. *PyEt* is meant as a community
project and the Authors welcome contributions and feedback to continue to improve and develop the project are welcome.

*Author contributions.* Conceptualization, M.V., S.B. and R.C.; software, M.V. and R.C.; investigation, M.V.; writing—original draft preparation, M.V. and R.C. ; writing—review and editing, S.B. and R.C.; supervision, S.B.. All authors have read and agreed to the published version of the manuscript.

*Competing interests.* The authors declare no conflict of interest.



*Acknowledgements.* We acknowledge the financial support by the University of Graz and the funding of the Earth System Sciences research
program of the the Austrian Academy of Sciences (ÖAW project ClimGrassHydro). We acknowledge the ZAMG dataset (https://data.hub.
zamg.ac.at), KNMI dataset (https://www.knmi.nl/home), and the E-OBS dataset from the EU-FP6 project UERRA (http://www.uerra.eu)
and the Copernicus Climate Change Service, and the data providers in the ECA&D project (https://www.ecad.eu).





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
