# Peer review of "Technical note: Improved handling of potential evapotranspiration in hydrological studies with *PyEt"

_Hydrology and Earth System Sciences, 2022_

## Author Comment (AC1)

We greatly appreciate the reviewers providing valuable and constructive comments on our manuscript HESS-2022-417. We considered each comment and will revise/improve the manuscript accordingly. The individual comments are replied below. In the following the reviewer comments are formatted in black font and our responses are blue.

Reply to RC1:

In this study, entitled 'Technical note: Improved handling of potential evapotranspiration in hydrological studies with PyEt', the authors present a new python-based software to calculate potential evaporation using a variety of semi-empirical equations. The paper is well-written and the package, in my opinion, is a useful addition to the variety of tools available for calculating hydrologic fluxes. I only have a few comments which can be addressed relatively easily.

Thank you for your encouraging and constructive comments. Your individual comments are replied to below.

1. A major issue I have is the suitability of HESS for this study. To me the study is the presentation of a software package and does not present 'new developments, significant advances, and novel aspects of experimental and theoretical methods and technique…' as required of a 'technical note'. It reads more like a GMD paper rather than a HESS paper.

While we acknowledge that our manuscript may also be suitable for publication in GMD, our main objective was not only to describe the software itself (which we consider more appropriate for GMD), but also to showcase a novel technique for estimating potential evapotranspiration (PET) using multiple PET methods for gridded datasets. We believe that by providing an environment in which users can estimate PET using up to 20 different PET methods, working with both time-series and gridded datasets (taking advantages of the xarray package), can be considered a new development in the field of PET estimation techniques. By publishing our manuscript and software in HESS, we aim to benefit the widest possible community in the fields of hydrology and earth system studies.

2. The name PyEt is misleading as the python package calculates potential evaporation rather than evaporation.

The P in PyEt stands for both Python and »Potential«.

3. The manuscript is missing details of compute times. I think this information is vital for any software package. How fast is the computation for gridded datasets? It would also be useful to have information about how efficient the software is regarding memory usage. I guess the use of xarray allows lazy loading and thus alleviates large memory usage.

In the revised manuscript, we will provide more detailed information on the compute times and memory usage for time series and gridded datasets.

4. Potential evaporation is not just used in hydrologic studies but in several ecological and climate-impact related studies. Therefore, the title does not completely do justice to potential use cases of the package.

We understand that the title of our manuscript may not fully reflect the potential use cases of the PyEt package. In fact, our manuscript is mainly aimed at hydrologists who need to calculate PET, which is why we try to illustrate and discuss the application of the package rather than the details of the technical implementation. Referring to comment no. 1, this is why we submitted the manuscript to HESS. Yet, we agree that the PET methods are also relevant in more general earth system studies

addressing ecological aspects or climate impacts. We will therefore change the title to: »Technical note: Improved handling of potential evapotranspiration in hydrological and earth system studies with PyEt«.

Line 65–70: 'hydrologists' is misspelled.

Line 2015: 'regions' should be 'region'.

We will correct the typos in the revised manuscript.

---

## Author Comment (AC2)

We greatly appreciate the reviewers providing valuable and constructive comments on our manuscript HESS-2022-417. We considered each comment and will revise/improve the manuscript accordingly. The individual comments are replied below. In the following the reviewer comments are formatted in black font and our responses are blue.

Reply on RC2:

The authors present a Python package that could help the community to implement evaporation equations. The package can easily compare differences between methods, intercalibrate models, and assess the effect of climate change by temperature and CO2. The package is open-source and is available in the now commonly used Python language. The manuscript is submitted as Technical note in HESSD.

Thank you for your feedback and constructive comments. Your individual comments are replied to below.

After checking the HESSD guidelines:*"Technical notes report new developments, significant advances, and novel aspects of experimental and theoretical methods and techniques which are relevant for scientific investigations within the journal scope. Manuscripts of this type should be short (a few pages only). Highly detailed and specific technical information such as computer programme code or user manuals can be included as electronic supplements. The manuscript title must start with "Technical note:". For manuscripts focused on the development and description of numerical models and model components, we recommend submission to the EGU interactive open-access journal Geoscientific Model Development (GMD).",* I would advice to move this manuscript to GMD.

While we acknowledge that our manuscript may also be suitable for publication in GMD, our main objective was not only to describe the software itself (which we consider more appropriate for GMD), but also to showcase a novel technique for estimating potential evapotranspiration (PET) using multiple PET methods for gridded datasets. We believe that by providing an environment in which users can estimate PET using up to 20 different PET methods, working with both time-series and gridded datasets (taking advantages of the xarray package) , can be considered a new development in the field of PET estimation techniques. By publishing our manuscript and software in HESS, we aim to benefit the widest possible community in the fields of hydrology and earth system studies.

Additionally, and more importantly, I doubt the advancement in science by this study. As said in Line 50, there already exist a similar package in R. So what is the added values of this work? Recoding from R to Python? I do realize that PyEt has some nice extra features, but in my view this is too little for a publication.

We regret that the manuscript does not effectively convey the importance of this work to the scientific community. It's worth noting that PyEt is not simply a Python copy of the R-package; there are several aspects of PyEt that go beyond that work, which we will highlight more effectively in the revised manuscript. These include: (1) PyEt provides direct computation of PET for 3D data, which, to the best of our knowledge, is directly not available in the existing R package. (2) PyEt can be easily integrated with other Python packages for sensitivity/uncertainty analyses, calibration, machine learning, etc. (3) The package is thoroughly tested with continuous integration and validated against literature values, which significantly reduces the chances of code errors. (4) As Python is one of the most popular programming languages, we target a much wider community.

Most importantly, the manuscript is not solely focused on the PyEt package, but presents its applications through a Jupyter Notebook, offering a step-by-step guide on estimating uncertainty in

PET estimations, computing PET for gridded datasets, calibrating PET models, estimating PET under climate change as well as a general discussion on the improved handling of PET in hydrological studies.

Furthermore, I do see some risks in this package. Of course, it the responsibility of modeller to select the right equation, and not of the developer; however, the current package seems not to have any disclaimers on the use and validity of certain models. At least, as how it is presented in the paper, the authors present the equations as interchangeable which is not correct. How can the user see that 'penman' and 'pm' are meant for different surfaces (open water, vegetated surface respectively)? How does the user know that Makkink is developed for Dutch landscapes (the factor 0,65 is a 'calibration' parameter)? I am rather sure other methods also have their limitations (e.g., local calibtation, model assumption, time scale). This is especially worrisome if I see Figure 3 where spatial patterns of PET are presented. I think PyEt should at least try to warn user for using proper formula's.

We appreciate your feedback on this matter and agree that describing the assumptions and limitations of each model is important. In the revised manuscript, we will add a table in the Appendix that outlines the key characteristics, assumptions, suitability, and limitations of each equation.

Moreover, I also wonder how the package deals with the different inputs. Solar radiation is rather easy to obtain, but how does the package deal with e.g., Penman (-Monteith)/FAO that requires net radation minus ground heat flux?

The package follows the guidelines and equations of the FAO-56 technical report to compute missing meteorological data for PET calculation. As PyEt computes PET on a daily scale, the default value for G is 0 (recommended in the FAO-56 paper), but users can input their own values for G. We will add a detailed table summarizing different inputs and methods for estimating internal meteorological variables in the Appendix.

Hence, to conclude: I appreciate the efforts of the authors to make the implementation of ET-models easier. I would have loved to have this package before, as I can't count the times how often I programmed certain formula's. However, I think the advancement in science it too little to merit publication in HESS.

While we understand that you may have some reservations about the scientific advancement presented in our manuscript, we believe that the package itself, the example applications, and the general recommendations derived from them represent significant contributions to the field of potential evapotranspiration estimation. By providing an open-source, user-friendly tool for researchers/hydrologists/geoscientist, we aim to improve the quality, efficiency, and reproducibility of work in these fields. We believe that the inclusion of new features such as direct computation of 3D potential evapotranspiration, and the ability to integrate the package with existing Python packages for sensitivity/uncertainty analyses, calibration, and machine learning, can be considered a new development in the field of PET estimation techniques . While these points were not specifically emphasized in the current manuscript, we will do our best to emphasize these aspects in the revised manuscript.

Minor comments:

- L21: it's funny that you prefer to use the term Evapotranspiration and cite in the same sentence the work of Miralles et al, 2020 where it is claimed that evaporation is the right term. Hence, I would advocate for call it 'potential evaporation'.

- Eq 2: ln should not be in italic

We will correct the errors that were identified in the revised manuscript.

- Eq 1&2: are these input parameter (LAI, CO2, zm, zom, d, zoh, etc) all fixed in the package and used for all equation?

The input parameters are not fixed, and users can provide their own values. However, if no input is given, default values based on FAO-56 guidelines are used. We will improve the description of this process in both the documentation and the Appendix.

- Example 4: this example is a nice add-on, but it's just on method of assessing the effect of climate change. So to me, it seems a bit a random choice (but not necessarily wrong btw).

Several studies highlighted the need to consider CO2 concentration when calculating PET for climate change related studies, typically with a modified Penman-Monteith equation. However, lack of meteorological data in certain regions or time periods can make using this equation impossible. We offer this example as a solution to this problem.